# Transcriptome Analysis Reveals the Molecular Mechanism of PLIN1 in Goose Hierarchical and Pre-Hierarchical Follicle Granulosa Cells

**DOI:** 10.3390/ani15020284

**Published:** 2025-01-20

**Authors:** Hua He, Yueyue Lin, Xi Zhang, Hengli Xie, Zhujun Wang, Shenqiang Hu, Liang Li, Hehe Liu, Chunchun Han, Lu Xia, Jiwei Hu, Jiwen Wang, Lin Liao, Xin Yuan

**Affiliations:** 1Farm Animal Genetic Resources Exploration and Innovation Key Laboratory of Sichuan Province, College of Animal Science and Technology, Sichuan Agricultural University, Chengdu 611130, China; hehua443@163.com (H.H.); lyy3078539326@163.com (Y.L.);; 2College of Horticulture, Sichuan Agricultural University, Chengdu 611130, China

**Keywords:** *PLIN1*, GCs, transcriptome, molecular mechanism, goose

## Abstract

The low laying performance of geese has significantly hindered the development of the goose breeding industry. The growth and development of goose follicle granulosa cells (GCs) directly influence normal ovulation in geese, which in turn impacts egg production. Our previous data suggest that de novo lipogenesis (DNL) occurs within the GCs of goose follicles. In this study, we investigated the potential mechanism of action of the *PLIN1* gene in goose GCs for the first time, using cell culture, transfection, and transcriptome sequencing technologies. The results of the transcriptome analysis indicated that *PLIN1* may regulate oxidative stress in goose follicular granulosa cells through the TGF-β signaling pathway, thereby inducing cell proliferation or apoptosis. Additionally, nine candidate genes were identified as potentially involved in this process. These findings enhance the understanding of the molecular regulatory mechanisms in goose follicular GCs and provide a theoretical foundation for future in-depth studies on these molecular mechanisms.

## 1. Introduction

As one of the most abundant proteins on the surface of lipid droplets of adipocytes, *PLIN1* (*PLIN1*) regulates lipid metabolism by regulating the contact between lipase and neutral lipids in lipid droplets. Relevant studies in mice, drosophila, and other model animals have confirmed that *PLIN1* can regulate lipid metabolism through the interaction with *HSL*, *CGI-58*, *ATGL,* and other lipolysis enzymes [1,2]. Data from related studies in chickens, pigs, and cows also show that *PLIN1* plays the same function of regulating lipid metabolism as in humans and mammals [3,4,5]. Data from our research team on goose follicle granulosa cells indicate that granulosa cells have the ability to de novo synthesize fatty acids and store neutral lipids in the form of intracellular lipid droplets. As the follicles develop, the lipid content of granulosa cells gradually increases, and *PLIN1* participates in the process of lipid deposition [6,7,8].

We hypothesize that when *PLIN1* regulation is disrupted, it will cause damage or even apoptosis of goose follicle granulosa cells. Research has confirmed that disturbances in intracellular lipid metabolism may lead to damage to some organelles, such as endoplasmic reticulum stress and changes in mitochondrial activity [9,10]. Mitochondrial damage can lead to an increase in intracellular ROS levels, which is necessary to trigger granulosa cell apoptosis. The excessive production of ROS can lead to an imbalance in the cellular oxidative antioxidant system, resulting in oxidative stress. Cellular oxidative stress can disrupt certain signaling pathways that control cell proliferation, autophagy, and apoptosis, leading to granulosa cell apoptosis [11]. In addition, some research data in vitro also indicate that the absence of *PLIN1* in liposarcoma cells can inhibit the proliferation and migration of liposarcoma cells by inducing G1 phase cell cycle arrest and apoptosis, and the absence in chicken preadipocytes can promote the proliferation of chicken preadipocytes [3,12]. In summary, we can speculate that if *PLIN1* is disorderly regulated in goose follicular granulosa cells, it is likely to cause damage or even apoptosis of goose follicular granulosa cells.

The role of *PLIN1* in other types of cells has been described, but there is no research revealing the mechanism of action of *PLIN1* on goose granulosa cells at present. Therefore, this study uses transcriptome sequencing technology to preliminarily explore the role of *PLIN1* in goose granulosa cells at the transcriptome level, as well as the regulatory pathways of *PLIN1*, especially whether there are differences in the regulation of hGCs and phGCs.

## 2. Materials and Methods

### 2.1. Animals and Sample Collection

A maternal line of Tianfu geese was raised under natural temperature and light conditions at the experimental station of waterfowl breeding at Sichuan Agricultural University. For follicle collection, twenty-four geese showing regular laying schedules were randomly selected; all were sacrificed 2 h after oviposition via post-anesthesia exsanguination. A pool of follicles was immediately collected from six goose abdominal cavities and placed in sterile normal saline. A pool of ovarian follicles was immediately collected from the goose abdominal cavities and divided into pre-hierarchal follicles and hierarchal preovulatory follicles according to previously reported nomenclature [13]. The granulosa layer was isolated as previously described [14]. All experimental procedures that involved animal manipulation were approved by the Committee of the School of Farm Animal Genetic Resources Exploration and Innovation Key Laboratory, College of Animal Science and Technology, Sichuan Agricultural University, under permit no. DKY20170913, and were performed in accordance with the Regulations for the Administration of Affairs Concerning Experimental Animals (China 1988). All efforts were made to minimize animal suffering conducted in accordance with the requirements of the Beijing Animal Welfare Committee.

### 2.2. Cloning CDS Region of Goose PLIN1

The mRNA reference sequence of the *PLIN1* gene was obtained from the reference genome of Sichuan white geese (GooseV1.0, GCF_002166845.1). The primers for amplifying the full-length CDS region were designed with the application of Primer Premier 5.0 software. We cloned the complete CDS region in three segments and then spliced the three cloned segments together. The primer sequences are shown in Table 1. Total RNA was extracted from the granulosa layer using TRIzol^®^ reagent (Invitrogen, Carlsbad, CA, USA) according to the manufacturer’s instructions, and genomic DNA was removed using DNase I (TaKaRa, Dalian, China). The quality of the extracted RNA was assessed using the 2100 Bioanalyzer (Agilent Technologies, Santa Clara, CA, USA), and its concentration was determined using the ND-2000 (NanoDrop Technologies, Wilmington, DC, USA). Only RNA samples meeting the following criteria were used for cDNA synthesis: OD260/280 ≈ 1.8–2.2, OD260/230 > 2.0, and RIN > 6.5. cDNA was synthesized using the HiScript III RT SuperMix for qPCR (+gDNA wiper) (Vazyme, Nanjing, China). PCR amplification was performed using cDNA as the template with the following reaction mixture: 1 μL forward primer (10 μmol/L), 1 μL reverse primer (10 μmol/L), 1 μL cDNA, 12.5 μL 2 × RapidTaq Master Mix, and 9.5 μL ddH_2_O. The PCR program was as follows: 95 °C for 3 min, followed by 35 cycles of 95 °C for 15 s, 60 °C for 15 s, and 72 °C for 27 s, with a final extension at 72 °C for 5 min. The amplified product was analyzed by agarose gel electrophoresis (1.5% agarose gel), and the target fragment was extracted and purified using the FastPure Gel DNA Extraction Mini Kit (Vazyme, Nanjing, China). The purified product was then ligated to a vector using the 5 min TA/Blunt-Zero Cloning Kit (Vazyme, Nanjing, China) and transformed into DH5α competent cells. The transformed cells were added to a liquid medium without ampicillin and cultured in a shaking incubator at 37 °C for 6 h. The culture was then centrifuged at 5000× *g* for 10 min. The supernatant (700 μL) was discarded, and the remaining liquid and precipitate were mixed and plated onto solid LB agar plates. The plates were incubated at 37 °C overnight. A single colony was selected and cultured in 1 mL LB medium containing 0.1% ampicillin at 37 °C for 4 h with shaking. The bacterial solution was then used as a template for PCR amplification using M13F (GTTGTAAAACGACGGCCAG) and M13R (CAGGAAACAGCTATGAC) primers. The PCR conditions and program were identical to those described above. The amplified products were sent to Sangon (Shanghai, China) for Sanger sequencing.

### 2.3. Construction of Overexpression Vectors and Synthesis of siRNA

We cloned the CDS region sequence of *PLIN1* (Appendix A), and the primers used for cloning are shown in Appendix A. The protein encoded by this gene was predicted using the online analysis tool NetPhos 3.1, provided by DTU Health Tech Bioinformatic Services (https://services.healthtech.dtu.dk/services/NetPhos-3.1/, accessed on 19 January 2023). The cloned *PLIN1* CDS region sequence was submitted to Sangon Biotech (Shanghai) Co., Ltd. (Shanghai, China), which synthesized the *PLIN1* overexpression vector (pcDNA3.1-*PLIN1*) and the negative control (pcDNA3.1(+)). Small interfering RNAs (siRNAs) targeting *PLIN1* mRNA were designed and chemically synthesized by GenePharma Co., Ltd. (Shanghai, China). The sequences of the siRNAs are provided in Table 2.

### 2.4. Primary Cell Culture and Transfection

Each time, 2–3 healthy female geese that were regularly laying eggs during the peak egg-laying period were selected. Euthanizing geese using the method of carbon dioxide and cervical dislocation induced dizziness; the abdominal cavity was opened, and the follicles were removed. They were placed in PBS buffer preheated at 37 °C and the hierarchical and pre-hierarchical granulosa layers were separated according to the operating steps. After washing with 1 × PBS (pH 7.3), they were placed in a 5 mL centrifuge tube, prepared with sterile scissors. The granulosa layers were completely cut, and then 0.1% type II collagenase (Sigma, Aldrich, St. Louis, MI, USA) was used, and the sample was shook while digesting until digestion was complete. Use DMEM/F12 medium containing 10% fetal bovine serum (DMEM/F12, Gibco, Shanghai, China) to terminate digestion. When preparing the medium, 50 mL of FBS was filtered using a 0.22 µM disposable filter, and then 45 mL of DMEM/F12, 5 mL of fetal bovine serum, and 500 uL of a penicillin/streptomycin mixture were added. After filtering through a 200 mesh filter, the cells were seeded on a 12-well culture plate with a density of 5 × 105 and placed in a 5% CO_2_, 37 °C incubator for cultivation. When the cells grew to 70–80%, they were transfected. Before transfection, the culture medium was replaced with DMEM/F12 medium containing 10% fetal bovine serum without dual antibodies. According to the manufacturer’s instructions, the *PLIN1* overexpression plasmid and siRNA (Table 1) were transfected with Lipofectamine 3000 (ThermoFisher Scientific, Carlsbad, CA, USA) for 24 h, and the cells were collected for RNA extraction.

### 2.5. Collection of Transcriptome Samples

After isolating and culturing both hierarchical and pre-hierarchical granulosa cells according to the previously described procedure, the cells were inoculated into 12-well plates. Once the cell growth reached 70–80%, the original medium was discarded, and DMEM/F12 medium containing 10% fetal bovine serum without double antibodies was added. The siRNA-PLIN1 and siRNA-NC negative control dry powders synthesized by Jima Gene Co., Ltd. were briefly centrifuged and then dissolved in the corresponding volume of DEPC water according to the instructions to prepare a 20 μmol/L stock solution, which was stored at −20 °C for future use. After extracting the pcDNA3.1-PLIN1 and pcDNA3.1 plasmids, they were stored in a −20 °C freezer for later use. In a sterile 1.5 mL EP tube, 20 μmol/L small RNA stock solution was diluted to 150 nmol/L using DMEM/F12 medium, labeled as solution A. The corresponding volume of plasmid DNA was added according to the mass and similarly diluted with DMEM/F12 medium. An equal volume of Lipofectamine 3000 was added based on the amount of small RNA and plasmid DNA in solution A and then diluted with DMEM/F12 medium, and the volume was adjusted to match the volume of solution A. The mixture was gently vortexed and labeled as solution B. Solutions A and B were then transferred to the same EP tube, gently mixed, and allowed to sit at room temperature for 10 min. Finally, the prepared transfection solution was slowly added to the cells. After gently shaking the culture plate to mix the transfection solution, the cells were placed in a 37 °C incubator for further culture, with at least three technical replicates for each treatment.

### 2.6. RNA Extraction and Sequencing

This study utilized a total of 24 samples, divided into 4 groups: phGC: over_vs_over NC; hGC: over_vs_over NCs; phGC: si_vs-_NC; hGC: si_vs_NC. Each group was further divided into a treatment group (over and si) and a control group (over-NC and si-NC), with 3 replicates used for both the treatment and control groups. Total RNA was extracted from 24 cell samples using Trizol reagent (Invitrogen, Carlsbad, CA, USA) and treated with DNase I (Invitrogen, Carlsbad, CA, USA) according to the manufacturer’s instructions. The RNA concentration and integrity were assessed using a NanoDrop 2000 spectrophotometer (Thermo Fisher Scientific, Waltham, MA, USA) and an Agilent 2100 Bioanalyzer system (Agilent Technologies, Palo Alto, CA, USA). We utilized the structural feature of most eukaryotic mRNAs possessing polyA tails to enrich these mRNAs using oligo(dT) magnetic beads. Fragmented mRNAs served as templates for synthesizing the first strand of cDNA in the M-MuLV reverse transcriptase system, with random oligonucleotides as primers. Subsequently, the RNA strand was degraded using RNase H, and the second strand of cDNA was synthesized using dNTPs in the DNA polymerase I system. After purification, the double-stranded cDNA underwent end repair, the addition of an A-tail, and sequencing adapter ligation. cDNA fragments approximately 250–300 bp in length were selected using AMPure XP beads, amplified by PCR, and purified again with AMPure XP beads to generate the final library. After quality assessment of the library, RNA sequencing was performed on the NovaSeq™ X Plus platform.

### 2.7. Data Analysis of RNA Sequencing

The raw data obtained from sequencing were filtered to remove reads with splices, containing N (where N represents base information that cannot be determined), and low-quality reads (Qphred ≤ 5 base number accounts for more than 50% of the entire read length). High-quality reads were located on the reference gene set using HISAT2 2.1.0, and uniquely located reads were assembled and quantified using StringTie v1.3.3, based on the FPKM (Transcripts Per Million) of each mRNA. For evaluating the value of gene expression, DEGs were identified using DESeq2 based on reading count data. *p*-value < 0.05 and |log2FoldChange| > 0.0 were used as critical values to screen for significantly differentially expressed genes. The online website KOBAS 3.0 (http://bioinfo.org/kobas, accessed on 12 January 2025) Gene Ontology (GO) and Kyoto Encyclopedia of Genes and Genomes (KEGG) functional analysis were used.

### 2.8. PPI Network Analysis

In this study, we conducted the PPI network interaction analysis using the STRING online platform (https://cn.string-db.org/, accessed on 19 January 2024). The minimum interaction score was set to 0.400 (medium confidence), and the other parameters were left at the default setting. PPI network mapping using Cytoscape software (version 3.5.1).

### 2.9. qRT-PCR Validation

Twelve DEG genes (*BAMBI*, *CCND1*, *PTEN*, *SMAD4*, *MYH11*, *SALL1*, *FOXP2*, *FST*, *JUN*, *FSHR*, *FOS*, and *SNX16*) were selected for qRT-PCR validation by using qRT-PCR technology. The reaction system consists of 10 µL, 1 µL cDNA, 5 µL 2 × ChamQ SYBR qPCRMaster Mix (Vazyme, Nanjing, China), 3.6 µL ddH_2_O, and 0.2 µL of each gene-specific primer (10 µM). Each sample was repeated 3 times and normalized to GAPDH using the 2^−ΔΔCt^ method [15], with the control set to one. Table 3 summarizes the primers for qRT-PCR.

### 2.10. Statistical Analyses

Results were presented as the mean ± SD from three independent experiments in triplicate. The differences were considered to be significant at *p* < 0.05. All experimental data were analyzed by ANOVA or nonparametric tests according to their test results of homogeneity of variance. IBM SPSS Statistic (version 27, Chicago, IL, USA) was used for all statistical analyses.

## 3. Results

### 3.1. Identification of Overexpression and Interference Efficiency

To detect the transfection efficiency of plasmids and siRNA, we simultaneously transfected siRNA-NC and siRNA-PLIN1 with fluorescent sequences into cells. After 12 h of transfection, the real-time fluorescence quantitative PCR technology was used to detect the expression of the target gene *PLIN1* after overexpression interference. The results showed that after overexpression of *PLIN1*, the *PLIN1* gene was overexpressed 401 times in granulosa cells (Figure 1A,B). Among the three siRNAs we designed, siRNA-PLIN1-476 has the best interference efficiency in hGCs and phGCs (Figure 1C,D).

### 3.2. Overview of RNA-seq and Identification of the DEGs

A total of 1,103,757,314 raw reads were obtained from the 24 samples, and an average of 45,604,314 clean reads were obtained from each sample after the strict screening. The Q20, Q30, and mapping rates were within 98.06% to 98.37%, 94.40% to 95.26%, and 86.06% to 89.25%, respectively, and the GC content was between 50.79% and 52.25% (Appendix A). Sequencing quality was acceptable for subsequent analysis.

With a *p*-value < 0.05 and |log_2_FoldChange| > 0.0 as the screening criteria, in phGCs, 918 DEGs were identified between over and over-NC, of which 382 were upregulated and 543 were downregulated; in hGCs, a total of 853 DEGs were identified between over and over-NC, of which 423 were upregulated and 434 were downregulated. Meanwhile, in phGCs, 1415 DEGs were identified between si and si-NC, of which 641 were upregulated and 780 were downregulated; in hGCs, a total of 917 DEGs were identified between si and si-NC, of which 469 were upregulated and 452 were downregulated (Figure 2).

### 3.3. Functional Enrichment Analysis of the DEGs

After overexpression and interference of PLIN1 in hGCs and phGCs, the DEGs obtained were subjected to GO and KEGG enrichment analyses. This experiment analyzed the differential expression of differentially expressed genes on three GO terms: biological processes (BPs), cellular components (CC), and molecular functions (MF). Select the most significant 30 terms and draw a bar chart for display (*p* < 0.05). If there are fewer than 30 terms, draw all terms (Figure 3A–D). Most DEGs were mainly enriched in the biological. Specifically, those related to cellular oxidative stress (response to oxidative stress, cellular response to oxidative stress), proliferation differentiation and apoptosis (apoptotic signaling pathway, positive regulation of cell death, regulation of cell differentiation), lipid metabolism and transport (negative regulation of cellular catabolic process, membrane lipid metabolic process, regulation of lipid transport), and protein phosphorylation (regulation of phosphorylation, regulation of protein phosphorylation) were significantly enriched (Table 4).

Based on the results of KEGG analysis, we selected the 20 most significant KEGG pathways from the 4 groups and plotted a scatter plot. If there were fewer than 20 pathways, all pathways were plotted (*p* < 0.05) (Figure 4A–D). The pathways with significantly enriched DEGs in PLIN1-overexpressed hGCs and phGCs include the TGF-beta signaling pathway, the Fanconi anemia pathway, and the adherens junction. This suggests that overexpression of *PLIN1* may affect the conduction of the TGF-β signaling pathway in hGCs and phGCs. In addition, nine signaling pathways were significantly enriched only in hGCs, including cell cycle, mitophagy—animal, autophagy—animal, and so on. However, 12 signaling pathways were significantly enriched in phGCs, including the mTOR signaling pathway, Wnt signaling pathway, protein processing in endoplasmic reticulum, and so on. Interference with *PLIN1* significantly enriched 22 signaling pathways in both hGCs and phGCs, including metabolic pathways, apoptosis, autophagy—animal, the MAPK signaling pathway, and so on. Nine signaling pathways were significantly enriched only in hGCs, including steroid biosynthesis, fatty acid biosynthesis, glycosaminoglycan degradation, and so on. Twenty-one KEGG signaling pathways were significantly enriched only in phGCs, including the TGF-β signaling pathway, cell cycle, fatty acid metabolism, fatty acid elongation, and so on. This indicates that overexpression or interference of *PLIN1* may have different effects on hGCs and phGCs.

### 3.4. Functional Enrichment and Network Analysis of Common Differentially Expressed Genes (co-DEGs)

In order to further explore the detailed molecular mechanism of *PLIN1* regulating lipolysis, we conducted Venn analysis on all DEGs in four groups. The results showed that the four groups had eight co-DEGs, namely *MGLL*, *BAMBI*, *ITA5*, *PPGB*, *EPAS1*, *ACTB*, *MSX2,* and *PCD10* (Figure 5A). Subsequently, KEGG enrichment analysis was performed for these eight genes.ss *BAMBI* is significantly enriched in the TGF-beta signaling pathway and Wnt signaling pathway; *MGLL* is significantly enriched in the glycerolipid metabolism; ACTB is significantly enriched in eight pathways, including Salmonella infection, adherens junction, apoptosis, phagosome, influenza A, tight junction, focal adhesion, and regulation of actin cytoskeleton (*p* < 0.05). Veen analysis of pathways significantly enriched with upregulated and downregulated genes revealed that in hGCs and phGCs overexpressing *PLIN1*, upregulated and downregulated DEGs were significantly enriched in the TGF-β signaling pathway (Figure 5B,D). Further PPI analysis of all DEGs in this pathway suggests that genes such as *FST*, *BAMBI*, *TGFB3*, and *BMPR2* may play a key regulatory role in this pathway, and the *BAMBI* gene is also a co-DEG among the four groups (Figure 5C). *BAMBI* was upregulated in both PLIN1-overexpressed hGCs and phGCs. *FST* was upregulated in both PLIN1-overexpressed and PLIN1-interfered phGCs. However, overexpression of *PLIN1* in hGCs resulted in downregulation of *BMPR2*, while in phGCs it resulted in upregulation of *BMPR2*.

### 3.5. Network Analysis of Upregulated and Downregulated DEGs

In addition, through KEGG analysis of downregulated and upregulated DEGs in four groups (Appendix A), we found that in hGCs and phGCs, PLIN1 interference significantly enriched five pathways in the upregulated DEGs, including focal adhesion, metabolic pathways, ErbB signaling pathway, cell apoptosis, and mTOR signaling pathway are significantly enriched in 19 downregulated DEGs pathways, which are related to lipid metabolism (metabolic pathways FoxO signaling pathway MTOR signaling pathway, autophagy (animal autophagy), proliferation and apoptosis (MAPK signaling pathway) are related. We also found that DEGs in phGCs downregulated between si and si-NC were dramatically enriched in KEGG pathways such as Fatty acid metabolism, Glycerolipid metabolism and Fatty acid biosynthesis. Subsequently, we constructed a PPI network diagram of all downregulated DEGs in hGCs between over and over-NC (Figure 6A). Through the analysis of the network, the genes with the highest number of nodes were found to include *PPARγ*, *WRN*, *ATR*, *MSH4* and *SMAD2Z*. Furthermore, we conducted PPI interaction network analysis on the DEGs upregulated between si and si-NC (Figure 6B). The results show that the top 9 DEGs with the highest number of nodes include *PTEN*, *JUN*, *NFKBIA*, *RACK1*, *UBL7*, *GPX1*, *DNAJC10*, *MYO9A* and *PAK1*. The key genes identified during the above analysis process are summarized in Appendix A.

### 3.6. Comprehensive Analysis of the Possible Molecular Mechanism of PLIN1 in hGCs and phGCs

We plotted the possible molecular mechanisms into a mechanism diagram (Figure 7). Overall, RNA-Seq analysis revealed that overexpression of *PLIN1* upregulated the expression of the *BAMBI* gene. The expression of the *BMPR2* gene is downregulated in hGCs and upregulated in phGCs, and further analysis revealed that both genes belong to the TGF β family, and their binding to *BMPR1* receptors will determine the activation or non-activation of apoptosis and cycle arrest-related pathways in this signaling pathway. Simply put, when *BMPR2* is upregulated in phGCs, its binding to *BMPR1* increases, leading to a decrease in *BAMBI* binding to *BMPR1*, while its binding to *TGFβRI* increases, inhibiting the apoptotic pathway in the TGFβ pathway and thus suppressing granulosa cell apoptosis. When *BMPR2* is downregulated in hGCs, its binding to *BMP1* decreases, leading to an increase in binding between *BAMBI* and *BMPR1*, while its binding to *TGFβRI* decreases, reducing its inhibitory effect and activating the apoptotic pathway, promoting granulosa cell apoptosis.

### 3.7. Verification of DEGs Expression Patterns

To verify that the change in DEGs was indeed caused by overexpression and interference of *PLIN1*, we selected 12 DEGs for qRT-PCR validation. The results are shown in Figure 8.

## 4. Discussion

In this study, we first cloned the *PLIN1* gene from the ovaries of Tianfu meat geese and then used phosphorylation prediction tools to identify the potential phosphorylation sites on the *PLIN1* protein. We found that *PLIN1* contains seven PKA phosphorylation sites. Phosphorylation of *PLIN1* is essential for optimal PKA-stimulated lipolysis, especially crucial for the translocation of *HSL* [16]. So, we speculated that these 7 phosphorylation sites of *PLIN1* may be related to the protein–protein interactions of *PLIN1* during lipid storage and hydrolysis.

To elucidate the possible molecular mechanism of *PLIN1* in goose granulosa cells, we overexpressed and interfered with *PLIN1* in hGCs and phGCs, respectively, and then sequenced and analyzed the transcriptome of the extracted RNA. The results of GO showed that most DEGs were enriched in BP functionally related pathways, indicating that BP may be more important for lipid metabolism in granulosa cells. The DEGs in phGCs between over and over-NC were significantly enriched in GO terms connected to cellular oxidative stress (response to oxidative stress and cellular response to oxidative stress). Oxidative stress is an imbalance between the oxidative and antioxidant systems caused by internal and external environmental stress factors and is one of the important reasons for abnormal function and apoptosis of granulosa cells [11,17]. Therefore, we speculate that overexpression of *PLIN1* may cause oxidative stress in granulosa cells. Our KEGG analysis suggests that overexpression or interference of *PLIN1* may have different effects on hGCs and phGCs. This point is also traceable; our research group’s preliminary data has shown that the lipid deposition ability of goose egg follicle granulosa cells varies at different stages [6]. We believe this is the reason for the difference in transcriptome levels between the two stages of cells.

In order to identify some common regulatory mechanisms of *PLIN1* in hGCs and phGCs, we performed Venn analysis on DEGs of four groups. The results of Venn’s analysis showed that there were eight co-DEGs in the four groups. When conducting KEGG analysis on these eight genes, we focused on two genes—*MGLL* and *BAMBI*. *MGLL* is significantly enriched in the glycerolipid metabolism. Monoacylglyceroesterase (*MGLL*) is an important metabolic enzyme that converts monoacylglycerols into free fatty acids and glycerol. It plays an important role in lipid metabolism and is associated with tumor-related signaling pathways, promoting the proliferation, invasion, and migration of UM cells [18,19]. In the TG hydrolysis process involving *HSL*, *ATGL*, and *MGLL*, *MGLL* hydrolysis is the final step. It can be said that *MGLL* can control the level of FFA in cells. The above evidence indicates that *MGLL* may also play a similar role in goose granulosa cells. Also, the *BAMBI* gene was found to be significantly enriched in the TGF beta signaling pathway and Wnt signaling pathway. Previous studies have shown that the Wnt signaling pathway works in conjunction with follicle-stimulating hormone (FSH) to regulate follicular development in humans, rats, mice, and pigs [20,21,22,23]. *BAMBI* is a transmembrane glycoprotein called TGF-β pseudoreceptors. It is upregulated in both PLIN1-overexpressed and PLIN1-interfered GCs and can regulate many biological phenomena, including glucose and lipid metabolism, inflammatory response, and cell proliferation and differentiation [24]. Previous studies have shown that BAMBI can regulate the proliferation of sheep follicular granulosa cells and the synthesis of steroid hormones [25]. Studies on pig granulosa cells have shown that *BAMBI* promotes the involvement of TGF-β signal transduction of steroid production in pig granulosa cells [26]. Therefore, we supposed that the *BAMBI* gene may also play a role in goose granulosa cell proliferation and steroid hormone production through the TGF beta signaling pathway.

Interestingly, the TGF-β signaling pathway is also a pathway that is significantly enriched in both upregulated and downregulated genes in follicular granulosa cells at both the pre-hierarchical and hierarchical stages after *PLIN1* overexpression. The TGF-β signaling pathway in mammals has pleiotropic functions, which have pleiotropic effects on cell proliferation, differentiation, adhesion, aging, and apoptosis [27]. Subsequently, we conducted PPI analysis on all the genes enriched in this pathway. The results indicate that genes such as *FST*, *BAMBI*, and *BMPR2* play a crucial role in the transmission of this signaling pathway. Research evidence suggests that *FST* is a stress response protein that plays a protective role under various stresses [28]. It prevents thapsigargin (Tg)-induced cell apoptosis by neutralizing ROS and subsequently inhibiting oxidative stress, which can be observed both in vitro and in vivo [29]. According to research data, *FST* knockout promotes the proliferation, migration, and invasion of BT549 and HS578T cell lines and inhibits mesenchymal cell apoptosis by targeting *BMP7* [30]. This indicates that after overexpression of *PLIN1* in goose granulosa cells, *FST* may play a role in oxidative stress and proliferation apoptosis of granulosa cells through the TGF-β signaling pathway. Some studies also indicate that the *FST* gene can also regulate the maturation of buffalo oocytes [31]. *BAMBI* is a group of four coDEGs, as mentioned earlier. It is interesting that the *BMPR2* gene is downregulated in PLIN1-overexpressed hGCs while it is upregulated in PLIN1-overexpressed phGCs. *BMPR2* is also a member of the TGF-β superfamily. Research data on adipocytes have shown that knocking out *BMPR2* is prone to cell death, and most importantly, this result is formed by inhibiting the phosphorylation of *PLIN1*, which affects fatty acid oxidation [32].

KEGG analysis was also performed for the upregulated and downregulated DEGs in hGCs and phGCs within the interference group. Focal adhesion, metabolic pathways, the ErbB signaling pathway, apoptosis, and the mTOR signaling pathway were significantly enriched among the upregulated DEGs. Previous studies have reported that the mTOR signaling pathway is associated with the apoptosis of chicken follicular granulosa cells [33,34]. This indicates that this pathway may also play a certain role in the apoptosis process of goose granulosa cells. Furthermore, downregulated DEGs were significantly enriched in 19 signaling pathways. All in all, these pathways are mainly related to cellular lipid metabolism and apoptosis autophagy. Our results also demonstrated that the downregulated DEGs in the pre-hierarchical group were significantly enriched in three pathways related to lipid metabolism, including fatty acid metabolism, glyceroid metabolism, and fatty acid biosynthesis. *ACSL4* and *ACSF3* are two key genes in the aforementioned pathways. *PLIN1*, *ACSF3*, and *ACSL4* are all genes in the PPAR signaling pathway. *ACSF3* is a mitochondrial enzyme that produces malonyl CoA from the toxic anti-metabolite malonic acid. Therefore, *ACSF3* performs a crucial metabolic editing function, allowing highly metabolically active cells to continue breathing [35]. *ACSL4* is a ferroptosis-related gene, and studies in pigs have shown that knocking out *ACSL4* reduces lipid droplet deposition [36].

PPI network analysis further revealed that, among the DEGs downregulated in the overexpression group at the hierarchical stage, *PPARG* had the highest number of nodes. *PPARG*, also known as *PPAR γ*, is enriched in the PPAR signaling pathway. It is a member of the ligand-activated transcription factor family, which can regulate adipocyte differentiation, lipid storage, and insulin sensitivity [37]. As the target genes of *PPARG*, PLIN1’s transcriptional regulation is highly related to the expression and activation of PPARG [38,39,40]. This indicates that the expression of *PLIN1* is also regulated by *PPARG* in goose granulosa cells. Research on chickens has also proven that the *PLIN1* is the target gene of *PPARG*; *PPARG* has a positive regulatory effect on the transcription of *PLIN1* in chickens [41]. In addition, our PPI results also showed that *PTEN* and *JUN* had the highest number of nodes among the upregulated DEGs in the interference group at the hierarchical stage. Studies on chickens have shown that *PTEN* can serve as a target gene for miR-22-3p and miRNA-29-3p. miRNA targeting PTEN activates the PI3K/Akt/mTOR pathway, promoting GC proliferation, steroid production, and lipid accumulation, regulating follicular development [42,43]. The Jun transcription factor family is one of the members of activator protein-1 (AP-1), which includes *JunB*, *c-Jun*, *JunD*, and *v-Jun*. It participates in the occurrence and development of various cancer syndromes by activating or inhibiting the transcription of multiple target genes. In recent years, studies have found that the Jun transcription factor family is involved in regulating various cellular functions such as proliferation, migration, and apoptosis of liver cancer cells [44]. It may weaken estrogen biosynthesis by directly downregulating the transcription of aromatase genes in ovarian granulosa cells [45]. In addition, reports have found that *JUN* is activated by MEK signaling in response to endoplasmic reticulum stress, and *JUN* binds to the promoters of several key UPR effectors (such as *XBP1* and *ATF4*) to activate their transcription and allow AML cells to correctly regulate endoplasmic reticulum stress [46]. Our KEGG results also showed that *JUN* is mainly enriched in pathways related to apoptosis and autophagy, such as the ErbB signaling pathway, apoptosis, the MAPK signaling pathway, and mitophagy. Based on existing reports, we speculate that after interfering with *PLIN1* at the hierarchical stage, the *JUN* gene may participate in the process of cell apoptosis by regulating cellular oxidative stress.

## 5. Conclusions

In conclusion, transcriptome sequencing identified 925 (phGC: over-vs_over NC), 857 (hGC: over vs_over NCs), 1421 (phGC: si-vs_i-NC), and 921 (phG: si-vs_si-NC) DEGs, respectively. Analysis of these DEGs screened a total of nine key genes (*PPARγ*, *MGLL*, *PTEN*, *BAMBI*, *BMPR2*, *JUN*, *FST*, *ACSF3*, and *ACSL4*) and one significant signaling pathway (TGF-β signaling pathway). Also, our study demonstrates that *PLIN1* plays a critical role in regulating granulosa cell function through the TGF-β signaling pathway, providing new insights into the molecular mechanisms underlying follicular development. This research is the first to establish a direct link between *PLIN1* expression and TGF-β pathway activation in granulosa cells, advancing our understanding of reproductive biology. However, this study was limited to the transcriptome level, and phenotypic validation at the cellular molecular level is needed to verify these findings. In future research, we will focus on the role of *PLIN1* in lipid metabolism, steroid hormone synthesis, proliferation and apoptosis, as well as oxidative stress in granulosa cells. In addition, the interaction between *PLIN1* protein and other lipid droplet proteins will also be a focus of future research.

## Figures and Tables

**Figure 1 animals-15-00284-f001:**
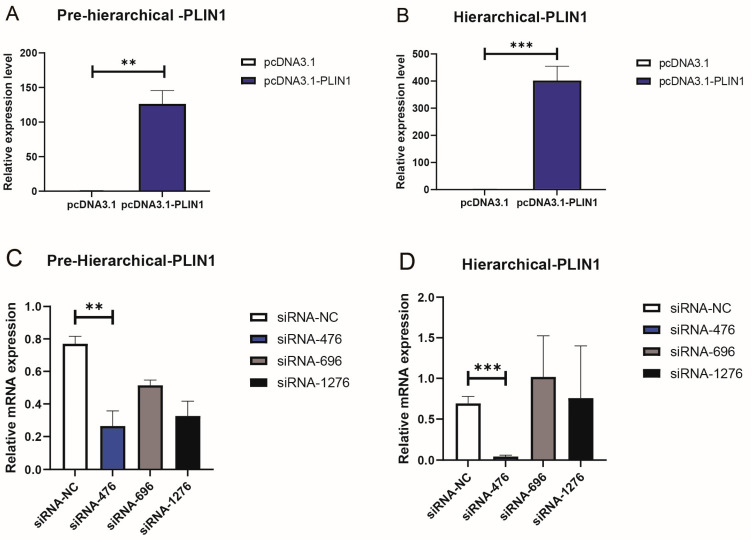
Identification of PLIN1 overexpression or interference efficiency. (**A**) Expression level of PLIN1 in hGCs after interference with PLIN1. (**B**) Expression level of PLIN1 in hGCs after overexpression of PLIN1. (**C**) Expression level of PLIN1 in phGCs after interference with PLIN1. (**D**) Expression level of PLIN1 in phGCs after overexpression of PLIN1. ** *p* < 0.01, *** *p* < 0.001.

**Figure 2 animals-15-00284-f002:**
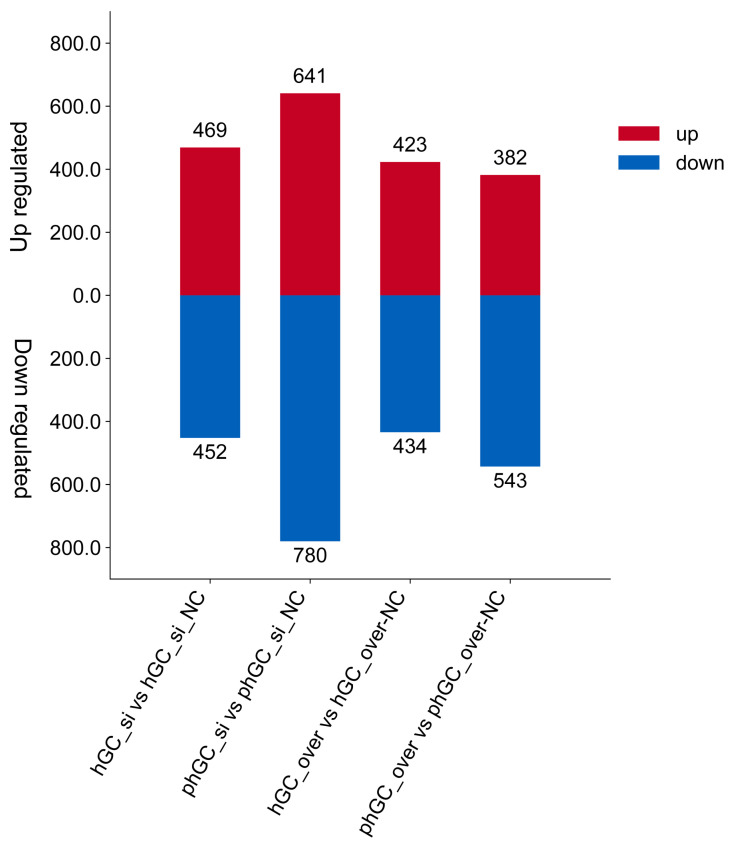
Identification of DEGs. Statistical analysis of the number of DEGs upregulated and downregulated by overexpression or interference in hGCs and phGCs.

**Figure 3 animals-15-00284-f003:**
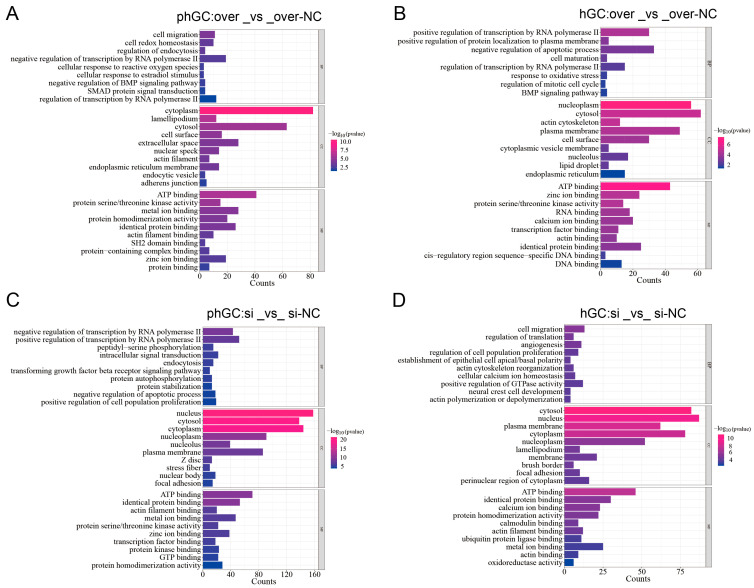
GO analysis of DEGs. (**A**) GO analysis of DEGs in phGCs of overexpression groups. (**B**) GO analysis of DEGs in hGCs of overexpression groups. (**C**) GO analysis of DEGs in phGCs of interference groups. (**D**) GO analysis of DEGs in phGCs of interference groups.

**Figure 4 animals-15-00284-f004:**
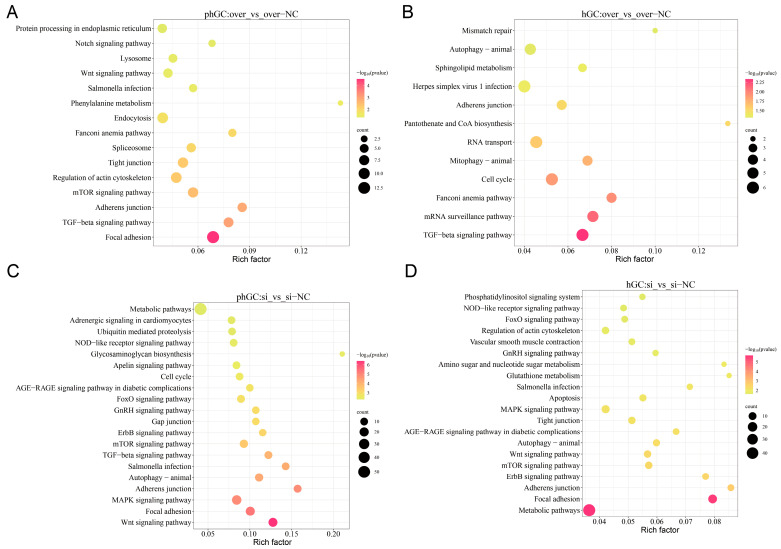
KEGG analysis of DEGs. (**A**) KEGG analysis of DEGs in phGCs of overexpression groups. (**B**) KEGG analysis of DEGs in hGCs of overexpression groups. (**C**) KEGG analysis of DEGs in phGCs of interference groups. (**D**) KEGG analysis of DEGs in hGCs of interference groups.

**Figure 5 animals-15-00284-f005:**
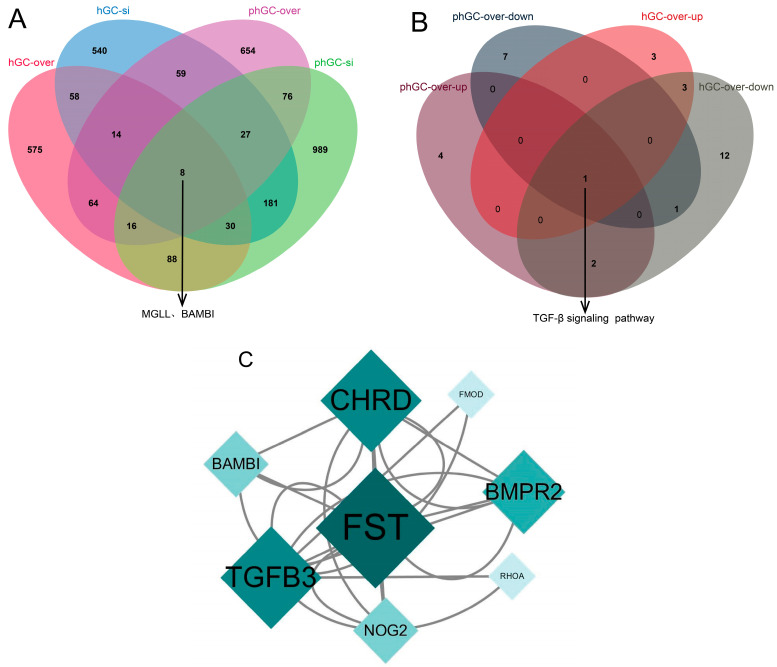
Analysis of common differentially expressed genes and pathways. (**A**) Venn analysis screening for common differentially expressed genes in 4 groups. (**B**) Venn analysis screens pathways enriched by differentially expressed genes upregulated and downregulated after PLIN1 overexpression in hGCs and phGCs. (**C**) PPI network diagram of all differentially expressed genes enriched in the TGF-β signaling pathway. (**D**) TGF-β signaling pathway.

**Figure 6 animals-15-00284-f006:**
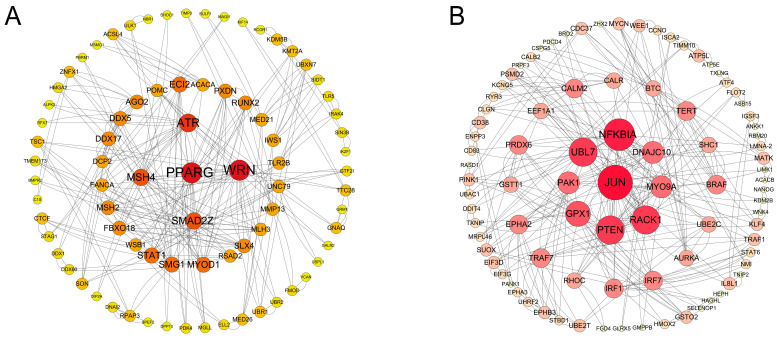
PPI network analysis of differentially expressed genes upregulated and downregulated by overexpression interference in hGCs. (**A**) PPI network analysis of differentially expressed genes upregulated by PLIN1 overexpression in hGCs. (**B**) PPI network analysis of differentially expressed genes downregulated by PLIN1 interference in hGCs.

**Figure 7 animals-15-00284-f007:**
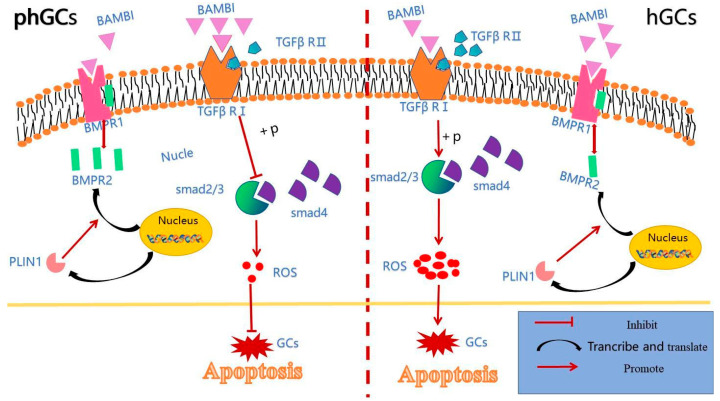
Mechanism diagram of PLIN1-induced cell proliferation and apoptosis in GCs.

**Figure 8 animals-15-00284-f008:**
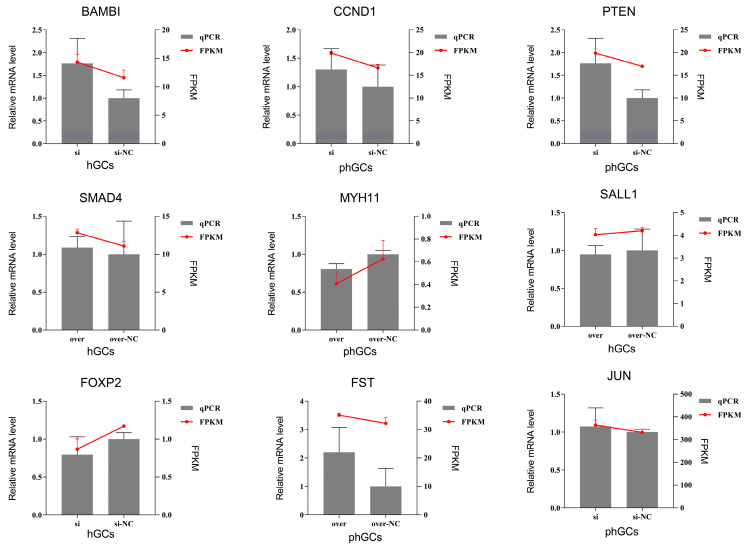
Verification of DEG expression patterns.

**Table 1 animals-15-00284-t001:** Primers for cloning PLIN1 gene.

Primer Name	Sense Primer	Anti-Sense Primer	Tm (°C)	Product Length (bp)
PLIN1-1	GTACAGCTACGTGCCGCTCC	TCAGTCCTTGAAGGCCG	67	788
PLIN1-2	CAGCAATGGTGGCGGAGGG	GCGTAGGTCCGCTGGAGGCT	60	355
PLIN1-3	TGTCGCGATGACGGCCAAGAAGAAT	GGTTGTAGGGGGCGAAGGCGGATTT	60	1493

**Table 2 animals-15-00284-t002:** PLIN1 siRNAs.

	Sense (5′–3′)	Anti-Sense (5′–3′)
siRNA-476 siRNA-696 siRNA-1276	CAGUUCACUGUAGCCAACATT CAGGUACACGAGGAGCAACTT AUCUCCAGCGUGAAGAAGGTT	UGUUGGCUACAGUGAACUGGG GUUGCUCCUCGUGUACCUGTT CCUUCUUCACGCUGGAGAUTT
siRNA-NC	UUCUCCGAACGUGUCACGUTT	ACGUGACACGUUCGGAGAATT

**Table 3 animals-15-00284-t003:** Primers for qRT-PCR.

Gene Name	Primer (5′-3′)	Tm (°C)	Product Size (bp)
BAMBI-F	ACTCACGGCTGCTTGGACTC	60	258
BAMBI-R	TGCCCTGAACCATAATTCTTTT		
CCND1-F	AGGAGCAGAAGTGCGAAGA	60	80
CCND1-R	TGCGGTCAGAGGAATAGTTT		
PTEN-F	TTGAAGACCATAACCCACCA	60	234
PTEN-R	CATTACACCAGTTCGTCCCT		
SMAD4-F	AGCAAGTGCGTTACAATACAG	60	212
SMAD4-R	GGCGATACTACACGCTCATA		
MYH11-F	GACCTGGTGGTGGACTTAG	55	287
MYH11-R	TTCTTGCCGACATCATCTT		
SALL1-F	AAAATTCACCAATGCCGTAG	60	237
SALL1-R	GACAAGCTGTCTTGCGATGC		
FOXP2-F	CATCACCACCAATAACTCATC	55	159
FOXP2-R	AATCTTCACAAACGCTTTCA		
FST-F	GCAAAGAAACGTGCGAGAAC	60	262
FST-R	GTGGAGCTGCCTGGACATAA		
JUN-F	AGAGAATCAAAGCCGAGCGA	60	156
JUN-R	CTCTGAGCATGTTGGCAGTG		
FSHR-F	AATGGAACCTACCTGGATGA	55	284
FSHR-R	GAGCAAGCCACATTAACGAC		
FOS-F	GCCCACCCTCATCTCCTCCG	60	222
FOS-R	GCCGCTGCCATCTTGTTCCTC		
SNX16-F	TATTGCTAACTGCCTTGC	54	234
SNX16-R	CCTGAAGATAAATCCCTAAT		
^1^ GAPDH-F	AGCAACATCAAGTGGGCAGA	60	157
^1^ GAPDH-R	CACCCATCACGAACATGGGA		

Abbreviations: F = forward primer; R = reversed primer; ^1^ house-keeping gene for data normalization.

**Table 4 animals-15-00284-t004:** Key GO terms.

ID	GO Name	*p*-Value
GO:0006979	response to oxidative stress	0.016831399
GO:0034599	cellular response to oxidative stress	0.027843103
GO:0097190	apoptotic signaling pathway	0.019373558
GO:0012501	programmed cell death	0.048648436
GO:0010942	positive regulation of cell death	0.037050845
GO:0045595	regulation of cell differentiation	0.026924001
GO:0031330	negative regulation of cellular catabolic process	0.040254762
GO:0032368	regulation of lipid transport	0.04245672
GO:0031325	positive regulation of cellular metabolic process	0.02113527

## Data Availability

The raw sequencing data are available in the SRA database (http://www.ncbi.nlm.nih.gov/sra (accessed on 25 September 2024) with the accession number PRJNA1181599.

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
