# Peer review of "Transcriptome Analysis Reveals the Molecular Mechanism of PLIN1 in Goose Hierarchical and Pre-Hierarchical Follicle Granulosa Cells"

_animals, 2025, doi:10.3390/ani15020284_

Round 1

Reviewer 1 Report (Previous Reviewer 1)

Comments and Suggestions for Authors

Thank you for revising the paper. I still think that the authors should consider changing the experimental conditions, but I understand that this research requires an environment that involves cell proliferation. I look forward to seeing further developments in future research.

Author Response

Responds to the reviewer 1 ’s comments:

Comments 1 : Thank you for revising the paper. I still think that the authors should consider changing the experimental conditions, but I understand that this research requires an environment that involves cell proliferation. I look forward to seeing further developments in future research.

Response: We firstly thank you for your valuable suggestions. As you mentioned, the current experimental design requires an environment conducive to cell proliferation, which is essential for our study's objectives. However, we recognize that exploring alternative conditions could offer valuable insights for future research. Currently, we are continuing this research, and the subsequent research data will be presented in another paper.We are committed to further optimizing our experimental approaches and will consider your recommendation for future studies.

Reviewer 2 Report (New Reviewer)

Comments and Suggestions for Authors

see attachment

Author Response

Responds to the reviewer2’s comments:

Comments 1 :Line 198: The significance parameters "p-value < 0.05" and "|log2FoldChange| > 0.0" may cause controversy. Could they be changed to more widely accepted parameters such as "q-value < 0.05" and "|log2FoldChange| > 1.0"?

Response: Thank you for your insightful comment regarding the significance parameters used in our study. We understand your concern about the potential controversy surrounding the use of "p-value < 0.05" and "|log2FoldChange| > 0.0." We chose these parameters because using them as screening criteria allows us to identify important genes and pathways, leading to more meaningful potential research directions. We will consider adopting these parameters in future studies to ensure that the results are more in line with common practices in transcriptomic research. We will revise the manuscript accordingly and clarify the rationale behind the parameter choices.

Comments 2:In the qPCR validation experiment, the expression levels of the two genes shown in the figure below are too low, which may cause controversy. Could you consider removing these two genes? Additionally, in line 196, the transcriptome expression quantification method is described as TPM, but in the qPCR figure, the label on the right side is FPKM, creating a conflict between the two.

Response: Thank you for the reviewer’s suggestion. Firstly, regarding the issue of low gene expression levels, from the trend in the figure, we can observe that although the gene expression is low, the quantitative expression trend of our gene still aligns with the transcriptome trend. This indirectly indicates the accuracy of our transcriptome sequencing data. Secondly, For the issue raised in line 196, we have made the necessary revisions and highlighted them. The change is as follows: Page 5,Line 196.

Comments 3:The gene names in the text need to be italicized, and in addition, the subheadings 3.1-3.7 in the results section are identical.

Response: Thank you for the reviewer’s suggestion. We have made the requested revisions one by one and highlighted them in red.

Comments 4:Some sentences in the manuscript need to be polished to enhance professionalism and fluency.

Response: Thank you for your valuable feedback. We will carefully review the manuscript and refine the sentences to improve their professionalism and fluency. Your suggestions are highly appreciated, and we will make the necessary revisions to ensure the manuscript meets the required standards.(Page 4,Line 162-166; Page 5,Line 203-204)

Comments 5:The description of cell transfection in the methodology section is insufficient and needs to be supplemented.

Response: Thank you for the reviewer’s suggestion. We have already made additions to this part. The change is as follows: Page 4,Line 166-181.

Comments 6:There are still some grammatical errors in the manuscript. Please check carefully. For example, in line 400, "group" should be plural.

Response: Thank you for carefully reviewing our manuscript. We have already made corrections to the grammatical errors. The change is as follows: Page 15,Line 400.

Comments 7:In Figure 8, the font size of phGC and hGC in the figure legend is too small, making it difficult for readers to view. Additionally, to maintain consistency with the text, "phGCs" and "hGCs" should be used instead of "phGC" and "hGC." Please make the necessary modifications.

Response: We firstly thank you for your valuable suggestions. Figure 8 has been modified accordingly based on the suggestions. Please refer to Figure 8 for details.

This manuscript is a resubmission of an earlier submission. The following is a list of the peer review reports and author responses from that submission.

Round 1

Reviewer 1 Report

Comments and Suggestions for Authors

The authors have investigated the function of the PLIN1 gene in geese granulosa cells using transfection and transcriptome analysis. Elucidating the role of PLIN1 in intracellular lipid metabolism would be significant, but I do not think that the experimental model used in this study is appropriate for this.

When considering the function of granulosa cells, it seems important to examine their response to gonadotropin stimulation and hormone secretion from granulosa cells, but these were not examined in this study, and it is not clear at what stage of granulosa cell development the authors were trying to observe the effects. Also, if the authors are trying to examine the response of granulosa cells to hormonal stimulation, it would be preferable to use an experimental system with a serum-free culture medium.

Although the cloning of PLIN1 itself is considered to be significant, this paper does not describe the details, and I do not evaluate this because I think it is different from the content that this paper is trying to assert.

Author Response

We would like to thank the Editor and Reviewers for their most helpful contributions and comments concerning our manuscript entitle “Transcriptome analysis reveals the molecular mechanism of PLIN1 in goose hierarchical and pre-hierarchical follicle granulosa cells’’. Those comments are very helpful for revising and improving our paper, as well as the important guiding significance to our researches. As per their suggestions, we have made all itemized changes and mark yellow color in paper. These changes did not influence the content or framework of the report. The main corrections in the paper and the responds to the comments are as flowing:

Responds to the reviewer 1 ’s comments:

Comments 1 :The authors have investigated the function of the PLIN1 gene in geese granulosa cells using transfection and transcriptome analysis. Elucidating the role of PLIN1 in intracellular lipid metabolism would be significant, but I do not think that the experimental model used in this study is appropriate for this.

Response: We thank you for your thoughtful comments. In this study, we used indigenous chinese tianfu geese as a research object which have low egg production rates and produce one egg every two days during an egg-laying period between September and May in the next year, the total number of eggs produced was about 80 in one egg-laying cycle. However, the poor laying performance of the goose is a hindrance to the industry. Hence, we have been working on the ovarian follicle research (Luo, Liu et al. 2018, Yuan, Deng et al. 2019, Zhang, Yao et al. 2019), hoping improving the laying performance of geese for increase economic value and ensure the future of the goose industry. Our previous studies provided the first evidence that fatty acid synthesis can occur in granulosa cells (Wen, Gan et al. 2019), and some related-studies (Gao, Gan et al. 2019, Li, Hu et al. 2019) have revealed that lipid metabolism had a vital function during follicular development. Results from our research group, including those by Ouyang Qingyuan et al., have highlighted the importance of lipid metabolism in egg production in geese. Excess lipid deposition in the body is a key factor contributing to low or absent egg production in geese.Currently, according to the developmental stages of follicles, follicles can be roughly divided into two stages based on size: pre-hierarchical follicles (6-10 mm) and hierarchical follicles (F5-F1). As the follicle develops, the lipid content in granulosa cells gradually increases, with lipid deposition being higher in hierarchical follicles than in pre-hierarchical ones. Moreover, the morphology of lipid droplets in granulosa cells changes as the follicle develops. The extent of lipid deposition in granulosa cells is related to their growth and development. Furthermore, the number of granulosa cells in the follicle determines whether follicular atresia occurs, which significantly reduces egg production in geese.

Previous studies by our research group have revealed the roles of certain genes related to lipid metabolism in the lipid metabolism, steroid hormone synthesis, and cell proliferation and apoptosis of granulosa cells in goose ovarian follicles. For example, the SCD gene has been studied in the context of lipid metabolism and steroidogenesis in goose granulosa cells [27, 28], while the ESR1 gene mediates lipid metabolism in granulosa cells at the dominant stage of follicular development, but not in earlier stages . Additionally, miR-202-5p inhibits lipid metabolism and steroidogenesis in goose granulosa cells by targeting ACSL3 , and targeting BTBD10 regulates granulosa cell proliferation and apoptosis to influence follicular selection . Therefore, further in-depth molecular-level studies on the relationship between lipid metabolism and egg production in geese are urgently needed. This project will be the first to investigate the role of lipid droplet-related protein (PLIN1) in lipid metabolism, proliferation, and apoptosis in goose granulosa cells, thereby further enriching the theoretical understanding of the molecular regulatory mechanisms of egg production in geese. Existing research data indicate that the PLIN1 gene is expressed in both steroidogenic cells and adipocytes, and goose granulosa cells are classified as steroidogenic cells. In adipocytes, PLIN1 interacts with kinases and esterases on the lipid droplet membrane to regulate the hydrolysis of stored triglycerides, thereby influencing the release of fatty acids and their transport to external cells. Additionally, the expression and function of PLIN1 are closely associated with the proliferation, differentiation, and regulation of lipid metabolism in adipocytes.The PLIN1 gene is a gene related to lipid metabolism function. Sequencing data from the early transcriptome of our research team showed that the PLIN1 gene is expressed in goose granulosa cells, differentially expressed before and after follicular selection, and ranked in the top 20 differential folds.

Comments 2 :When considering the function of granulosa cells, it seems important to examine their response to gonadotropin stimulation and hormone secretion from granulosa cells, but these were not examined in this study, and it is not clear at what stage of granulosa cell development the authors were trying to observe the effects. Also, if the authors are trying to examine the response of granulosa cells to hormonal stimulation, it would be preferable to use an experimental system with a serum-free culture medium.

Response: Thank you for the reviewer's comment. Hormone testing is currently being conducted, which is an important means of studying the subsequent functions of this part. This part of the data is being organized and planned to be published in another article. In this study, we aim to systematically reflect the potential function of the PLIN1 gene in goose granulosa cells using transcriptome sequencing data. Additionally, the study intends to observe two stages of granulosa cell development: pre-hierarchical follicular granulosa cells (phGC) and hierarchical follicular granulosa cells.In the process of follicular development in poultry (such as chickens), ovarian follicles can be categorized into hierarchical follicles and pre-hierarchical follicles based on their developmental stage and size. These two types of follicles exhibit significant differences in developmental stages, functions, and physiological characteristics. Pre-hierarchical follicles refer to follicles in the early developmental stages that have not yet entered the rapid growth phase or the hierarchical system. These follicles have a diameter of less than 10 mm, are numerous, and show minimal yolk deposition. The yolk accumulation has just begun, with low protein and lipid content. They appear milky white or pale yellow in contrast to the bright yellow of hierarchical follicles and are not yet ready for ovulation. Hierarchical follicles refer to follicles in the ovary that are in the rapid growth phase and nearing maturity. These follicles exhibit a distinct size hierarchy during development, arranged from largest to smallest, with significant yolk deposition. They are typically the follicles about to participate in ovulation. The follicles are ranked in size from largest to smallest (e.g., F1, F2, F3...F6), with F1 being the follicle closest to ovulation, followed by F2, F3, and so on. There are significant functional differences at the molecular level between graded follicles and pre-graded follicles. Therefore, most of the studies conducted by our team are based on these two stages.

Comments 3:Although the cloning of PLIN1 itself is considered to be significant, this paper does not describe the details, and I do not evaluate this because I think it is different from the content that this paper is trying to assert.

Response: Thank you for the reviewer’s suggestion. Following the recommendation, we have moved the content originally in the appendix into the main text. The changes are as follows:Page 2-3,Lin94-L126

Reviewer 2 Report

Comments and Suggestions for Authors

The study attempts to unveil the regulatory mechanism of PLIN1 in follicular granulosa cells (GCs) of geese by applying cell transfection and transcriptome sequencing methods. The authors overexpressed PLIN1 in hGCs and phGCs and utilized GO and KEGG for gene enrichment analysis to trace the TGFβ signaling pathway, the only significantly enriched pathway. A thorough gene expression and pathway analysis have been provided in the manuscript to understand PLIN1 overexpression and interference on GCs, highlighting specific siRNAs with superior interference efficiency. It contributes to the existing knowledge of reproduction.

Comments:

>> The specific protocols for RNA extraction and library construction in the methodology section are incomplete and need to be detailed keeping an eye on the reproducibility of the results.

>> There is no discussion on the limitations such as sample size constraints or potential biases. It may impact the results. Address it.

>> The discussion on the functional significance of the identified DEGs and their implications for GCs in reproductive processes can enhance the overall merits. Further, compare the findings with existing literature on PLIN1 and GCs in the broader contexts for their relevance in the reproductive processes in the ovary. It will also help in understanding the research outcomes.

>> The study is missing the validation of the outcomes. Additional experiments such as qRT-PCR or functional assays need to be performed to confirm the findings to consolidate the conclusions.

>> It may be valued if the authors add a section on potential future research directions. The scientific value of the study will be increased by identifying areas for further exploration or experiments to expand on the existing findings.

>> Conclusion needs to be rewritten. It should be compact and not messy, unclear, and assumption-based.

>>Line 179:  ‘Data are presented as standard deviations ±SD’. What is this?

>>  Line 185-186 Typographical errors- siRAN. There are many such visible errors. Check the manuscript thoroughly.

>> Revisit the manuscript for the English language for clarity and articulation along with some grammatical mistakes.

Comments on the Quality of English Language

The English language needs improvement. Many grammatical and typographical errors are present in the manuscript. 

Author Response

Responds to the reviewer ’s comments:

Comments 1:Thank you for the reviewer's comment. The specific protocols for RNA extraction and library construction in the methodology section are incomplete and need to be detailed keeping an eye on the reproducibility of the results.

Response: We used the Trizol lysis method to extract RNA, which is a conventional laboratory RNA extraction method. Due to the routine steps, we will not describe them in detail. Other content has been modified as required. (Page 4-5,Line 170-189).

Comments 2:There is no discussion on the limitations such as sample size constraints or potential biases. It may impact the results. Address it.

Response: Thank you for the reviewer's comment, it has been modified as suggested and the location of the change is as follows: Page 4,Line 171-173.

Comments 3:The discussion on the functional significance of the identified DEGs and their implications for GCs in reproductive processes can enhance the overall merits. Further, compare the findings with existing literature on PLIN1 and GCs in the broader contexts for their relevance in the reproductive processes in the ovary. It will also help in understanding the research outcomes.

Response: Thank you for the reviewer's comment .This is a highly professional and valuable opinion. We initially considered this issue, but after reviewing the literature, we found that there are currently no research reports related to PLIN1 and granulosa cells.

Comments 4:The study is missing the validation of the outcomes. Additional experiments such as qRT-PCR or functional assays need to be performed to confirm the findings to consolidate the conclusions.

Response: We firstly thank you for your valuable suggestions. Due to the lack of previous reports on PLIN1 and granulosa cells, our research aims to comprehensively reveal the function of PLIN1 in goose egg granulosa cells at the transcriptome level. We validated our sequencing results using qRT PCR technology(Page 14,Figure 8), and further functional validation will be our research direction in the future.

Comments 5: It may be valued if the authors add a section on potential future research directions. The scientific value of the study will be increased by identifying areas for further exploration or experiments to expand on the existing findings.

Response:We believe your suggestion is very valuable, and we will include this point in the conclusion(Page 17,Line 504-508).

Comments 6:Conclusion needs to be rewritten. It should be compact and not messy, unclear, and assumption-based.

Response: We appreciate your insightful suggestions. We have revised the conclusion section according to your suggestions. Please refer to the conclusion section of the revised version of the article(Page17,Line 493-508).

Comments 7: Line 179:  ‘Data are presented as standard deviations ±SD’. What is this?

Response: Thank you for the reviewer's comment, the intent of this section we would like to convey is that our results are presented as the mean ±standard deviation(SD). We have made changes in the article(Page 6,Line 218-223).

Comments 8: Line 185-186 Typographical errors- siRAN. There are many such visible errors. Check the manuscript thoroughly.

Response: Thank you for the reviewer's comment. We have corrected the errors, and we have also corrected those with the same errors, which have been highlighted in yellow.

Comments 9: Revisit the manuscript for the English language for clarity and articulation along with some grammatical mistakes.

Response: Thank you for the reviewer's comment. We have reviewed the grammar again and made corrections to the grammatical issues in the article. Please refer to the yellow-highlighted sections in the revised version.